# Metamorphic Integration of GaInAsSb Material on GaAs Substrates for Light Emitting Device Applications

**DOI:** 10.3390/ma12111743

**Published:** 2019-05-29

**Authors:** Qi Lu, Andrew Marshall, Anthony Krier

**Affiliations:** Physics Department, Lancaster University, Lancaster LA1 4YB, UK; a.r.marshall@lancaster.ac.uk (A.M.); a.krier@lancaster.ac.uk (A.K.)

**Keywords:** GaInAsSb, molecular beam epitaxy, interfacial misfit arrays, dislocation filtering

## Abstract

The GaInAsSb material has been conventionally grown on lattice-matched GaSb substrates. In this work, we transplanted this material onto the GaAs substrates in molecular beam epitaxy (MBE). The threading dislocations (TDs) originating from the large lattice mismatch were efficiently suppressed by a novel metamorphic buffer layer design, which included the interfacial misfit (IMF) arrays at the GaSb/GaAs interface and strained GaInSb/GaSb multi-quantum wells (MQWs) acting as dislocation filtering layers (DFLs). Cross-sectional transmission electron microscopy (TEM) images revealed that a large part of the dislocations was bonded on the GaAs/GaSb interface due to the IMF arrays, and the four repetitions of the DFL regions can block most of the remaining threading dislocations. Etch pit density (EPD) measurements indicated that the dislocation density in the GaInAsSb material on top of the buffer layer was reduced to the order of 10^6^ /cm^2^, which was among the lowest for this compound material grown on GaAs. The light emitting diodes (LEDs) based on the GaInAsSb P-N structures on GaAs exhibited strong electro-luminescence (EL) in the 2.0–2.5 µm range. The successful metamorphic growth of GaInAsSb on GaAs with low dislocation densities paved the way for the integration of various GaInAsSb based light emitting devices on the more cost-effective GaAs platform.

## 1. Introduction

With a wide tunable bandgap of 0.28–0.73 eV, the GaInAsSb compound material lattice-matched to GaSb has found various applications in photo-detectors [1], lasers [2] and thermophotovoltaic cells [3]. Most significantly, the GaInAsSb based quantum well lasers on GaSb substrates have exhibited superior performance in the mid-infrared spectral range [4,5]. The epitaxial growth of GaInAsSb has been predominantly carried out on GaSb substrates. High-quality GaInAsSb material on GaSb was reported from both MBE and metalorganic vapor-phase epitaxy (MOVPE) growths [6].

As an expensive material itself, the high cost of the GaSb substrates can potentially limit the applications of the GaInAsSb based devices. In contrast, the GaAs (001) substrates can serve as a cost-effective III-V semiconductor platform with mature processing techniques. Transplanting the GaInAsSb material onto GaAs substrates will not only help reduce the overall cost but also pave the route for the monolithic integration of GaInAsSb and GaSb based devices with the existing GaAs based opto-electronic systems. One of the major challenges for the growth of GaInAsSb on GaAs is the large lattice mismatch between the two materials, which will result in a high density of (threading dislocations) TDs arising from the interface during the direct epitaxy of GaInAsSb on GaAs. The TDs, which can act as non-recombination centers and electrical channels, can seriously deteriorate the material quality and device performances [7,8]. Various epitaxial methods have been investigated for the suppression of TDs during the growth of materials with large lattice mismatch, which include graded buffer layers [9], dislocation filtering layers [10], interlayer structures [11] and strain compensation [12]. However, little has been reported on the integration of GaInAsSb based devices on GaAs substrates.

In this work, we proposed a novel technique for the metamorphic growth of GaInAsSb material on GaAs substrates by MBE. The buffer layer on GaAs was initiated with the (inter-facial misfit) IMF arrays on the GaSb/GaAs interface and included four repetitions of GaInSb/GaSb (dislocation filtering layers) DFLs. The GaInAsSb material lattice-matched to GaSb was grown on top of the buffer region, and GaInAsSb P-N diodes were fabricated. The TDs in the buffer region was investigated by cross-sectional (transmission electron microscopy) TEM images. The TD density in the GaInAsSb region was estimated from (etch pit density) EPD measurements. Photo-luminescence (PL) from the material and electro-luminescence (EL) from the diode were measured from 6 K to room temperature. The low TD density and bright luminescence from this material grown on GaAs have made it possible to integrate the full structures of more complicated GaInAsSb and GaSb based light emitting devices onto the GaAs platform in the future.

## 2. Experiments 

The growth was carried out in a Veeco GENxplor solid source MBE system (Plainview, NY, USA) on 2-inch n-GaAs substrates. The substrates were baked at 350 °C in high vacuum for one hour before the growth. After the growth of about 200 nm GaAs material, GaSb was deposited on top with the IMF arrays at the GaSb/GaAs interface at a substrate temperature (T_s_) of 560 °C. One micrometer thick GaSb was grown after the T_s_ was lowered to 500 °C. Afterwards, four regions of DFLs were deposited at a T_s_ of 445 °C, each containing 10 periods of GaInSb/GaSb MQWs and separated by 270 nm GaSb region in between. The composition of the GaInAsSb lattice-matched to GaSb was calibrated in advance. With a bandgap of about 0.5 eV, the composition was estimated as Ga_0.78_In_0.22_As_0.20_ Sb_0.80_ [13]. The GaInAsSb P-N diode structure was grown on top of the DFL regions with 1 µm n-GaSb (doped to ~5 × 10^17^ cm^−3^), followed by 4 µm p-GaInAsSb (doped to ~2 × 10^17^ cm^−3^) and 100 nm thick GaSb contact layer heavily p-doped to ~2 × 10^18^ cm^−3^. The structure of the sample is illustrated in Figure 1. Prototype GaInAsSb LEDs were processed with conventional photolithography and wet etching techniques. Eight hundred micrometer diameter mesas were etched in an H_3_PO_4_ and C_6_H_8_O_7_ based solution. Afterwards, the sidewalls were treated in a solvent of HCl:H_2_O:H_2_O_2_ (100:100:1) to eliminate possible surface defects [14]. Ti (20 nm)/Au (200 nm) contacts were thermally evaporated on top of the structure, and GeAu (20 nm)/Au (200 nm) contacts were made by the same method on n-GaSb. The fabricated devices were mounted on TO headers for characterization.

The TEM specimens were prepared using standard grinding, polishing and Ar^+^ ion-milling to electron transparency, and examined in JEOL 2100 LaB6 TEM (Coventry, UK) operating at 200 kV. Geometric phase analysis (GPA) measurements were performed using the program Strain++ (V 1.5, University of Warwick, Coventry, UK), in which the GaAs substrates were set as the reference. EPD measurements were carried out by dipping the sample of GaInAsSb on GaAs into an HNO_3_ and citric acid based solution for 45 s at 20 °C. The TD density in the GaInAsSb region was then estimated by counting the etch pits numbers revealed on the optical microscopy images. High-resolution double crystal X-ray diffraction (XRD) scan with a Bede QC200 diffractometer (Durham, UK) was used to examine the structure of the DFL regions. The samples were mounted on a cold finger in a liquid helium cooled cryostat and excited by a 650 nm wavelength semiconductor laser with about 50 mW output power and ~1 mm diameter spot size. The PL spectra were obtained using a Fourier transform infrared (FTIR) spectroscopy set-up (Vertex 70, Bruker, Billerica, MA, USA) at different temperatures. The EL from the GaInAsSb diode grown on GaAs was measured using the same FTIR apparatus. Electric current was sent into the device from a pulse generator in 1 KHz frequency and 10% duty circle.

## 3. Results

### 3.1. Material Characterizations

The cross-sectional TEM images in Figure 2 revealed the behaviors of the TDs in the buffer region. As shown in Figure 2a, the IMF arrays at the GaSb/GaAs interface clearly suppressed a large percentage of the TDs from penetrating into the GaSb region. Figure 2e showed a high magnification TEM image of the GaSb/GaAs interface, where the IMF arrays (marked by the red arrows) can be clearly observed, similar to the work of Huang et al. [15]. The IMF technique can efficiently release the strain caused by the lattice mismatch. The GPA can directly reveal the strain behavior in the sample [16]. As shown in the GPA image of the interface in Figure 3a, the strain was highly localized at the position of the IMF arrays. And the GaSb material (upper half of the image) was uniform without dislocations in this area. However, the IMF arrays alone may not result in low enough TD densities for producing high-quality materials and devices, since some TDs can still arise from the interface. Figure 2b showed a section of the buffer region where more TDs emerged from the interface and penetrated through the 1 µm thick GaSb region. The GPA image of the TDs arising from the interface was also shown in Figure 3b, where localized strain in the GaSb buffer region can be observed, corresponding to the TDs. 

To further reduce the TD densities, four DFL regions were grown on top of the 1 µm GaSb. Each DFL region was composed of 10 periods of strained GaInSb/GaSb MQWs and separated by 270 nm thick GaSb in between, as can be observed in Figure 2a,b. The high-resolution TEM images of the top and bottom DFL regions are shown in Figure 2c,d respectively. Sharp interfaces between the GaInSb and GaSb materials can be clearly seen in both regions. In comparison, the top DFL region contained almost no TDs, while there were much higher densities of TDs around the bottom DFL region. The efficient reduction of TD densities by the DFL regions is also evident in Figure 2a,b, as we can observe almost dislocation-free material above the top DFL region.

After etching away the GaInAsSb P-N structure, XRD measurements were carried out on the DFL regions. Figure 4a shows the ω-2θ spectrum from the scan (black curve), compared with the simulated spectrum (red curve). The peak at the 0 position was from the GaAs substrate and the one at around −9570˝ came from the GaSb material. Clear shoulder peaks can be observed on both sides of the GaSb peak, which originated from the GaInSb/GaSb MQWs. The simulation indicated that in the structure, the GaInSb and GaSb layers were 10.1 nm and 11.5 nm in thickness respectively. In addition, the indium composition in the GaInSb layers was about 17%. The Ga_0.83_In_0.17_Sb/GaSb produced a strain of 1.1%. The shoulder peaks in the measured curve appeared to be weaker and broader compared with the simulated curve, which could possibly be due to the still existing TDs in the DFL regions. The density of the TDs in the GaInAsSb material above the DFL regions was estimated by the EPD measurements. The sample was dipped into a solution containing citric acid and nitric acid [17]. Due to the faster etching rates at the sites of dislocations, the location of the TDs on the planar surface can be revealed. Figure 4b shows the optical microscopy image of the sample after dipping into the solution. From this picture, by counting the number of square-shaped etch pits, the density of the TDs in the GaInAsSb region can be estimated to be in the 10^6^/cm^2^ range, which was among the lowest reported value for this type of material grown on GaAs.

### 3.2. Photo-Luminescence and Electro-Luminescence

The PL spectra of the GaInAsSb material grown on the GaAs substrates with the novel metamorphic buffer layer technique were displayed in Figure 5a. The peak position shifted from 2.26 µm at 6 K to 2.52 µm at 300 K due to the bandgap narrowing with increasing temperature. The linewidth of the spectra also broadened from 48 nm at 6 K to 196 nm at 300 K. The EL spectra of the GaInAsSb P-N diode on GaAs are also shown in Figure 5b when 100 mA pulsed current was injected into the device. Compared with the PL results, very similar trends in the shift of peak position and the broadening of the linewidth can be observed from the EL spectra. The peak moved from 2.25 µm to 2.53 µm, and the linewidth broadened from 68 nm to 231 nm when the temperature went up from 6 K to 300 K. However, the intensity of the EL signal showed much less reduction with rising temperature than the PL signal. The peak intensity of the PL spectrum at 300 K dropped to only about 1.4% of the value at 6 K. In contrast, the peak intensity of the EL spectrum at 300 K maintained 7% of that at 6 K. In addition, the GaInAsSb diode on GaAs showed very similar I-V characteristics under forward bias with the same structure grown on lattice-matched GaSb substrates. At −0.1 V reverse bias, the current density from the diode on GaAs was 5.6 mA/cm^2^, still higher than the 1.9 mA/cm^2^ from the diode on GaSb.

## 4. Discussion and Conclusions

It is highly desirable to directly grow lattice mismatched materials and structures on the more mature substrate platforms such as GaAs and Si for the cost reduction and integration of opto-electronic systems. Up to date, various material systems have been experimented for metamorphic integrations, including InAs on GaAs [18], GaSb on GaAs [19], InGaAsP on Si [20] and GaAs on Si [10]. The mismatch will result in a high density of TDs from the interface and penetrating in the device regions. Thus, careful engineering in the epitaxial growth technique is required to suppress the TD densities. Among the epitaxial techniques for lattice mismatched growth, the DFLs have been widely used to efficiently reduce the TD densities [21]. For example, DFLs have been successfully implemented in the MBE growth of various III-V photonic devices on Si substrates where they have achieved excellent performance [22,23]. The lowest TD density in the order of 10^5^/cm^2^ has been reported for the metamorphic integration of GaAs on Si. Besides the DFL technique, the IMF arrays have been developed particularly for the metamorphic growth of GaSb on GaAs. Subsequently, GaSb based lasers [24], LEDs [25] and detectors [26] on GaAs with IMF arrays have been reported. In this work, the IMF arrays were first deposited on the GaSb/GaAs interface, reducing the TD density to the order of 10^8^/cm^2^ near the interface. The four regions of GaInSb/GaSb DFLs were designed by following the rules suggested in the theoretical investigation by Ward et al. [27], which further suppressed the TD density to the order of 10^6^/cm^2^ in our sample, resulting in one of the lowest TD densities for the GaInAsSb grown on GaAs. Further reduction of the TD densities can possibly be achieved by adding in-situ annealing during the MBE growth of the DFL regions, as was used in the work of Chen et al. [10]. In addition, the successful metamorphic growth of the GaInAsSb material with a lattice constant of 6.09 Å on GaAs substrates can be utilized for the integration of 6.1 Å family materials (InAs, GaSb and AlSb) for various types of devices and more complicated structures.

In summary, we designed a novel metamorphic buffer layer approach in the MBE growth to integrate the GaInAsSb material on cost-effective GaAs substrates. The buffer region was composed of GaSb/GaAs IMF arrays and Ga_0.83_In_0.17_Sb/GaSb strained layers as the DFLs. The behavior of the TDs in the buffer region was revealed by TEM and GPA images, which showed effective suppression of the TDs by the IMF arrays on the interface and further significant reduction of the TD density by the DFLs. The XRD measurement indicated the strain in the DFLs was about 1.1%, close to the suggested value by the theoretical work. The EPD characterization revealed the TD density was in the order of 10^6^/cm^2^ in the GaInAsSb region, which was among the lowest for this material grown on GaAs. The GaInAsSb material showed bright PL at room temperature at around 2.5 µm. Finally, the LED based on the GaInAsSb P-N diode was fabricated, which exhibited strong EL at 300 K. The successful integration of the GaInAsSb on GaAs with the advanced metamorphic buffer layer paved the way for the epitaxial growth of more complicated light emitting devices based on the GaInAsSb and GaSb materials onto GaAs substrates.

## Figures and Tables

**Figure 1 materials-12-01743-f001:**
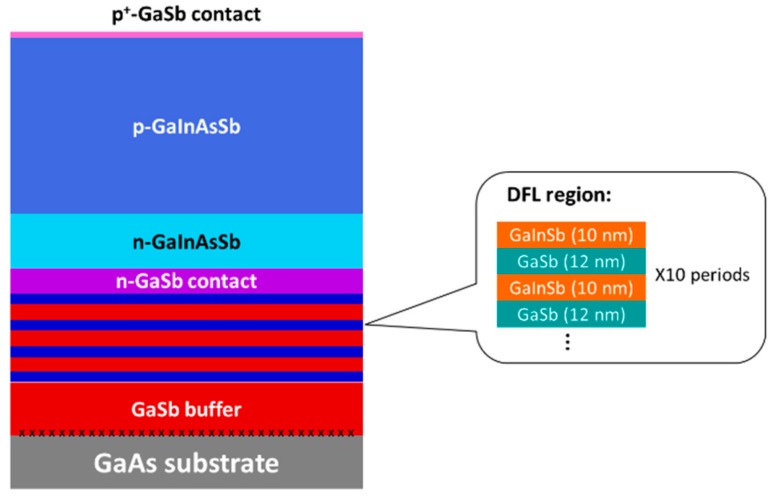
Schematic illustration of the GaInAsSb P-N diode grown on GaAs substrate with interfacial misfit (IMF) arrays and four dislocation filtering layer (DFL) regions. The details of the DFL regions are included in the inset.

**Figure 2 materials-12-01743-f002:**
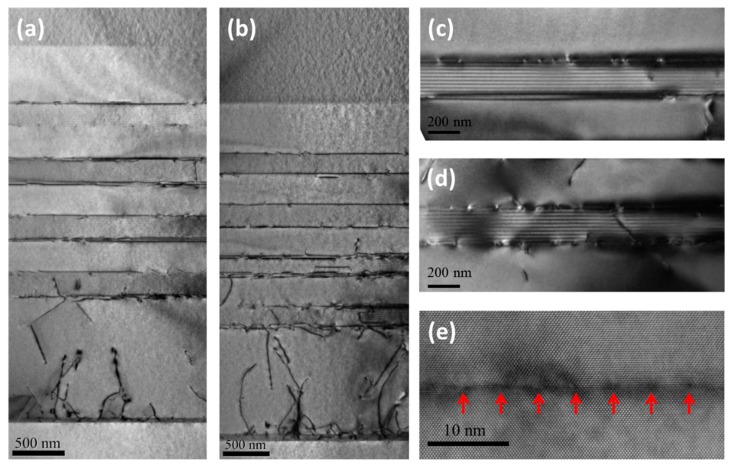
Cross-sectional TEM images of the metamorphic buffer region (**a**) with low TD density, and (**b**) with high TD density. High-resolution TEM images revealed the details of (**c**) the top DFL region, (**d**) the bottom DFL region and (**e**) the GaSb/GaAs interface.

**Figure 3 materials-12-01743-f003:**
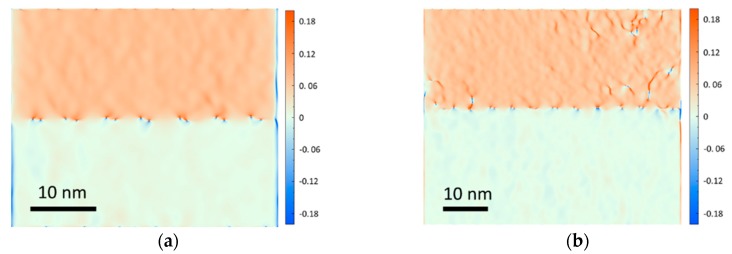
Geometric phase analysis (GPA) images of the GaSb/GaAs interface: (**a**) without TDs; (**b**) with TDs.

**Figure 4 materials-12-01743-f004:**
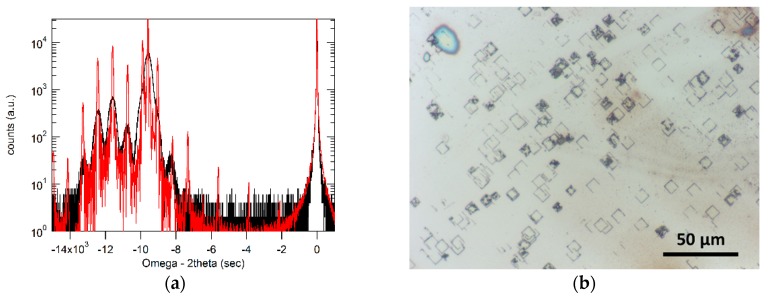
(**a**) measured (black curve) and simulated (red curve) XRD spectra of the GaInSb/GaSb DFL regions; (**b**) optical microscopy image of the sample after EPD measurements.

**Figure 5 materials-12-01743-f005:**
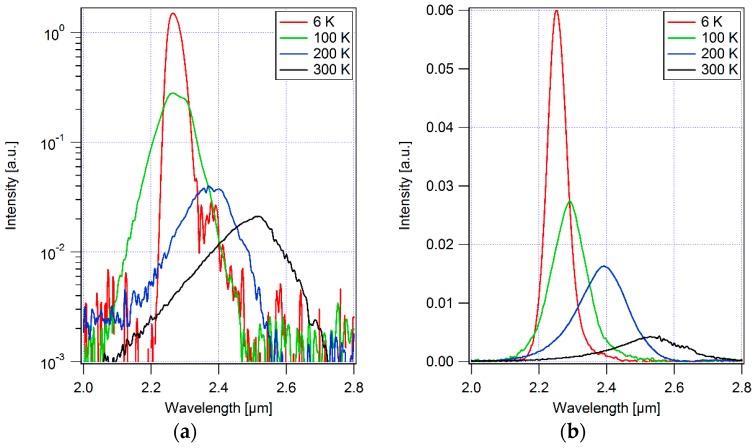
(**a**) Photo-luminescence spectra of the GaInAsSb material grown on GaAs at different temperatures; (**b**) electro-luminescence spectra of the GaInAsSb P-N diode grown on GaAs at different temperatures.

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
