# Peer review of "Metamorphic Integration of GaInAsSb Material on GaAs Substrates for Light Emitting Device Applications"

_materials, 2019, doi:10.3390/ma12111743_

Reviewer 1 Report

The article "Metamorphic integration of GaInAsSb material on GaAs substrates for light emitting device applications" describes the design of a specific buffer (multi)layer design to reduce threading dislocations.

The article describes a well-known mechanism albeit for a very specific case, where a taylored multilayer stack was designed. As such, the research is not original but interesting enough for the III-V community. The experimental results are well presented and the articles reads well. I recommend publication, and have only one minor comment that should become addressed.

- p 3 ll97-98 & fig. 3: For the GPA analysis it is important that the authors indicate the region of reference in the figure. Was it in the GaSb? The statement "The GaSb material was largely strain free" depends on where that region was chosen, i.e. if the region of reference for the phase maps was chosen to lie within the GaSb then it is no wonder that it appears as strain free since it would be referenced to itself. This is a crucial point here. Also, the parameters for the GPA analysis, accuracy and resolution (aperture size) should be stated for the reader to better interpret Fig. 3.

Author Response

We thank the reviewer for pointing out the confusion about the GPA measurements. The reference in the GPA measurements was the GaAs substrates. As shown in Fig 3(a), the lower half of the image (GaAs material) was mostly in white color (zero strain). And the red color in the upper half (GaSb material) was due to the different lattice constant of GaSb as opposed to GaAs.

We also realized that the GPA analysis in this case cannot strictly prove the GaSb is strain free. The uniform color in the GaSb region in Fig 3(a) only indicated the material is uniform and dislocation free. The comment on page 3 line 97-98 has been changed accordingly in the revised manuscript.

Reviewer 2 Report

The paper is interesting and worth to be published.

However, I'd like the authors to clarify/address the following points:

line 45: it may be worth, (especially for students) to cite the possibility of defect reduction also by strain compensation. See for example Megalini, L.; Šuran Brunelli, S.T.; Charles, W.O.; Taylor, A.; Isaac, B.; Bowers, J.E.; Klamkin, J. Strain-Compensated InGaAsP Superlattices for Defect Reduction of InP Grown on Exact-Oriented (001) Patterned Si Substrates by Metal Organic Chemical Vapor Deposition. Materials 201811, 337

line 58. Any cleaning procedure before loading the samples into the MBE?

line 70: how effective is the cleaning mentioned in reducing the defects? please quantify or add a reference

line 80: what is the temperature of the etch pit solution?

the defect density reported is ~10^6. How much is the reduction compared to the same epi layer grown w/o any defect engineering techniques?

line 128: please provide an estimation or cite a work indicating the difference in etch rate of the different crystal plane

line 155: as in the case of line 45. It is helpful, especially for students who will read this paper, that a big and hot topic is also InP grown on Si, in particular by MOCVD since this the standard technique in industry. Cite for example Megalini, L., Cabinian, B.C., Zhao, H. et al. Journal of Elec Materi (2018) 47: 982. https://doi.org/10.1007/s11664-017-5887-9

Is there any difference in electrical performance compared to a device not grown on GaAs, for example in Vf?

Author Response

Firstly, we thank the reviewer to provide additional reference works. The two papers by Megalini et al was referenced in line 45 in the revised article (reference [12]), and in line 160 (reference [20]) in the revised manuscript.

line 58. Any cleaning procedure before loading the samples into the MBE?

There was no additional cleaning before loading the substrates for the MBE growth. The substrates were baked at 350 degrees in high vacuum for 1 hour before the growth, which has been added in line 59 in the revised manuscript.

 line 70: how effective is the cleaning mentioned in reducing the defects? please quantify or add a reference.

We didn't carry out comparative study on the effectiveness of this cleaning method. The recipe was from the work of Juang et al, which has been added as reference [14] in the revised article. In their study, the sidewall leakage was greatly suppressed by using this cleaning method.

 line 80: what is the temperature of the etch pit solution?

The etch pit solution was at 20 degrees during the experiment. It was added in line 81 of the revised manuscript.

 the defect density reported is ~10^6. How much is the reduction compared to the same epi layer grown w/o any defect engineering techniques?

We haven't carried out detailed characterization of the material grown on GaAs without any metamorphic engineering techniques. Our preliminary study showed much wider XRD peaks and possible surface roughness if the GaInAsSb or GaSb was directly deposited on GaAs. The theoretical study by Ward et al (reference [27] in the revised manuscript) predicted the dislocation density could be in 1x10^7 to 1x10^8 /cm2 range by growing 2-3 micrometer buffer layer without any extra structure.

 line 128: please provide an estimation or cite a work indicating the difference in etch rate of the different crystal plane

The reference work for our EPD recipe has been added in the revised manuscript (reference [17]). The dislocations in semiconductor can serve as preferential etching sites, though it is difficult to quantify how much the etching rate changes. To be more accurate, we have also modified the sentence in line 128 from "due to the difference in etching rate in different lattice orientations" to "due to the faster etching rate at the dislocation sites."

 Is there any difference in electrical performance compared to a device not grown on GaAs, for example in Vf?

We have added some electrical comparison between the diode on GaAs and the same structure grown on GaSb. Under forward bias, the I-V curves are very similar. Under reverse bias, the diode on GaAs still showed relatively larger dark current. Please see the added texts in line 149-152 on page 5.